psychology

emotion detection, emotion recognition, facial expression, schizophrenia, schizotypy, visual search

**Author for correspondence:**
Shota Uono
e-mail: uono@ncnp.go.jp

†Present address: Department of Developmental Disorders, National Institute of Mental Health, National Center of Neurology and Psychiatry, 4-1-1 Ogawahigashi, Kodaira, Tokyo 187-8553, Japan.

# Schizotypy is associated with difficulties detecting emotional facial expressions

Shota Uono[1,†], Wataru Sato[3], Reiko Sawada[2,4], Sayaka Kawakami[2], Sayaka Yoshimura[1] and Motomi Toichi[2,4]

[1]Department of Neurodevelopmental Psychiatry, Habilitation, and Rehabilitation, Graduate School of Medicine, and [2]Faculty of Human Health Sciences, Graduate School of Medicine, Kyoto University, 53 Shogoin-Kawahara-cho, Sakyo-ku, Kyoto 606-8507, Japan
[3]Kokoro Research Center, Kyoto University, 46 Shimoadachi, Sakyo-ku, Kyoto 606-8501, Japan
[4]The Organization for Promoting Neurodevelopmental Disorder Research, 40 Shogoin-Sannocho, Sakyo-ku, Kyoto 606-8392, Japan

SU, 0000-0003-0547-0522

People with schizophrenia or subclinical schizotypal traits exhibit impaired recognition of facial expressions. However, it remains unclear whether the detection of emotional facial expressions is impaired in people with schizophrenia or high levels of schizotypy. The present study examined whether the detection of emotional facial expressions would be associated with schizotypy in a non-clinical population after controlling for the effects of IQ, age, and sex. Participants were asked to respond to whether all faces were the same as quickly and as accurately as possible following the presentation of angry or happy faces or their anti-expressions among crowds of neutral faces. Anti-expressions contain a degree of visual change that is equivalent to that of normal emotional facial expressions relative to neutral facial expressions and are recognized as neutral expressions. Normal expressions of anger and happiness were detected more rapidly and accurately than their anti-expressions. Additionally, the degree of overall schizotypy was negatively correlated with the effectiveness of detecting normal expressions versus anti-expressions. An emotion–recognition task revealed that the degree of positive schizotypy was negatively correlated with the accuracy of facial expression recognition. These results suggest that people with high levels of schizotypy experienced difficulties detecting and recognizing emotional facial expressions.

## 1. Introduction

Schizophrenia is a chronic and severe psychiatric condition characterized by positive (e.g. delusion and hallucination),

negative (e.g. flat affect and anhedonia), and/or disorganized (e.g. thought disorder) symptoms [1]. It has been proposed that schizophrenia is located at the extreme end of a spectrum of psychotic symptoms and that the opposite end includes a non-clinical general population with schizotypy [2,3]. Schizotypy represents a personality structure composed of psychological factors that are similar to those of schizophrenia in terms of positive, negative and disorganized symptom domains [4]. Previous studies have found that elevated levels of schizotypal traits are associated with the later emergence of schizophrenia spectrum disorder [5] and share core features with schizophrenia in the cognitive, neural and genetic domains [6]. The study of schizotypy allows us to investigate the psychological mechanisms associated with schizophrenia and related characteristics under cognitively demanding tasks, because schizotypal traits are distributed across the general population [7].

In addition to the core symptoms of schizophrenia, recent studies have placed considerable attention on the social impairments [6,8]. Schizophrenia is often accompanied by difficulties recognizing facial expressions [9]. Impairments in emotion–recognition exist prior to the onset of schizophrenia [10], are positively correlated with the severity of positive and negative symptoms [9], and represent a promising target for interventions that improve functional outcomes [11]. Similarly, several studies have shown that individuals with high schizotypal traits show impairments recognizing facial expressions and that the degree of impairment is positively correlated with the levels of schizotypy measure [6,12–15]. These findings reveal that the impairment in the processing of facial expressions is an associated feature of schizophrenia symptom and schizotypal traits. Given that the ability to process facial expressions is associated with functional outcomes in the clinical population [16,17], the findings emphasize the need for a better understanding of the psychological mechanisms underlying facial expression processing relevant to schizophrenia and schizotypy.

However, some studies have reported preserved perceptual and emotional processing of facial expressions in schizophrenia [18]. Behavioural studies have demonstrated that emotional facial expressions presented under a priming procedure [19–21] or continuous flash suppression [22] can bias subjective evaluation of subsequent neutral stimuli in schizophrenia and control groups. Furthermore, threatening faces enhanced the performance of a delayed matching task [23], and such expressions interfered with a gender discrimination task in schizophrenia and control groups [24]. These findings suggest that the perceptual and emotional aspects of facial expressions can be automatically processed in patients with schizophrenia and affect their behaviour in the same manner as in healthy individuals.

Neuroscience studies have raised a question about evidence that the perceptual and emotional processing of facial expressions are preserved. Electrophysiological and functional magnetic resonance imaging studies have demonstrated dysfunction in early visual processing of faces in patients with schizophrenia [25,26]. A meta-analysis revealed that patients with schizophrenia exhibit restricted brain activation in response to emotional facial expressions in the amygdala [27]. The processing of facial expressions includes several stages such as visual processing, emotional and motor response, and interpretation of their meanings [28]. The perceptual and emotional processing stage prior to recognition may play an important role in this impairment, because it can induce a cascading effect on subsequent processing. On the assumption that detecting emotional facial expressions in complex environments serves as a starting point for initiating meaningful interaction, we investigated whether the detection of emotional facial expression versus neutral expressions was associated with the levels of schizotypy in the general population.

Only a few studies have investigated the detection of emotional facial expressions using clinical populations and the findings are inconsistent. One study presented a schematic angry or happy face among neutral faces and found that both the patient and control groups detected angry faces more rapidly than happy faces, whereas the patient group showed a delayed reaction in general [29]. However, because this study did not compare the detection of emotional versus neutral faces, it remains unclear whether emotional faces were detected more rapidly than neutral faces. A recent study found that the accuracy of detecting a happy face among neutral faces is lower in people with schizophrenia than in controls [30], which suggest that patients have impairments when detecting happy faces. However, these authors observed that patients also had impairments when detecting indicated neutral faces and did not directly compare the detection of emotional and neutral faces. Thus, findings in this area remain inconsistent and unclear, but this may be, at least partially, due to differences in clinical conditions. Studies assessing an effect of schizotypy in general populations may be helpful for revealing whether the rapid detection of emotional versus neutral faces is impaired across the schizophrenia spectrum.

The present study examined whether differences in the detection of emotional facial expressions would be associated with the degree of schizotypy after statistically controlling for the effects of age,

sex and IQ. Participants were asked to answer whether all faces were the same as quickly and accurately as possible when angry and happy faces and their anti-expressions were presented within crowds of neutral faces. Anti-expressions produced by computer morphing are morphologically reversed faces of normal emotional facial expressions; they contain a degree of visual change in facial features (e.g. eyebrows and lips) equivalent to those of normal emotional facial expressions relative to neutral facial expressions and are recognized as relatively neutral expressions compared with normal emotional expressions [31]. This approach allows for comparisons of the detection of emotional versus emotionally neutral facial expressions while controlling for the amount of visual change in facial features. Thus, the enhanced detection of normal versus anti-expressions is attributable to the enhanced emotional processing rather than the visual difference between target faces. Previous studies of general populations have shown that normal expressions, especially anger expressions, are detected more rapidly than their anti-expressions [32–35]. Based on ample evidence showing relatively poor social cognition and impaired facial expression processing in people on the schizophrenia spectrum, it was hypothesized that the degree of schizotypy would be negatively correlated with the ability to more rapidly detect normal expressions versus anti-expressions. To confirm the lower emotional impact of anti-expressions versus normal expressions, the participants were asked to rate the emotional valence and arousal that was experienced for each target stimulus. Finally, the emotion–recognition test was used to replicate the finding that the impairment of explicit processing of facial expressions positively correlates with the degree of schizotypy.

# 2. Material and methods

## 2.1. Participants

The sample size was determined through *a priori* power analysis for multiple regression analysis using G*power [36] assuming the intermediate effect size ($r = 0.342$) obtained in the previous schizotypy study [37], $\alpha$ level of 0.05 and power of 0.8. The results showed that 62 participants would be needed. Considering the possibility that the RTs under the emotion–detection task cannot be calculated in some participants because of too low accuracy rate, a total of 70 Japanese adults (age: $M = 21.7$, s.d. = 2.2; 26 females and 44 males) were recruited from Kyoto University. The resulting sample size was four times larger than that of the original study investigating emotion–detection [32] and was similar to that of the study investigating the association with personality traits [34]. Full-scale IQ was estimated using four subtests of the Japanese version of the Wechsler Adult Intelligence Scale, third edition (WAIS-III); the digit symbol coding, matrix reasoning, digit span and information tasks. None of the participants had an intellectual disability ($M = 121.3$, s.d. = 12.9).

The participants completed the Japanese version of the Schizotypal Personality Questionnaire (SPQ) [38,39], which includes 74 items. The total SPQ score was calculated by adding up the 'yes' responses to the items, and scores on the nine subscales were used to assess the degree of each symptom domain based on a three-factor model: positive (ideas of reference, odd beliefs and magical thinking, unusual perceptual experience and suspiciousness) negative (suspiciousness, excessive social anxiety, no close friends and constricted affect) and disorganized schizotypy (odd and eccentric behaviour, and odd speech). The total score of the participants (total: $M = 27.6$, s.d. = 12.4; Positive: $M = 9.49$, s.d. = 5.88; negative: $M = 13.99$, s.d. = 7.09; disorganized: $M = 7.36$, s.d. = 3.93) was comparable to that of the large sample study of a Japanese population [38].

All participants provided written informed consent prior to participation. The experiment was approved by the Kyoto University Graduate School and Faculty of Medicine, Kyoto University Hospital Ethics Committee (R0658), and was conducted in accordance with Ethical Guidelines for Medical and Health Research Involving Human Subjects.

## 2.2. Apparatus

All tasks were controlled using Presentation (Neurobehavioural System; https://www.neurobs.com/) implemented on a Windows computer (HP xw4300 Workstation) and all stimuli were presented on a 17-inch CRT monitor (Iiyama; Tokyo, Japan) with a refresh rate of 100 Hz. The distance between the monitor and the participant was maintained at approximately 57 cm using a headrest. All responses were recorded using a response box (RB-530, Cedrus; San Pedro, CA) that measures reaction time (RT) with a 2–3 ms resolution.

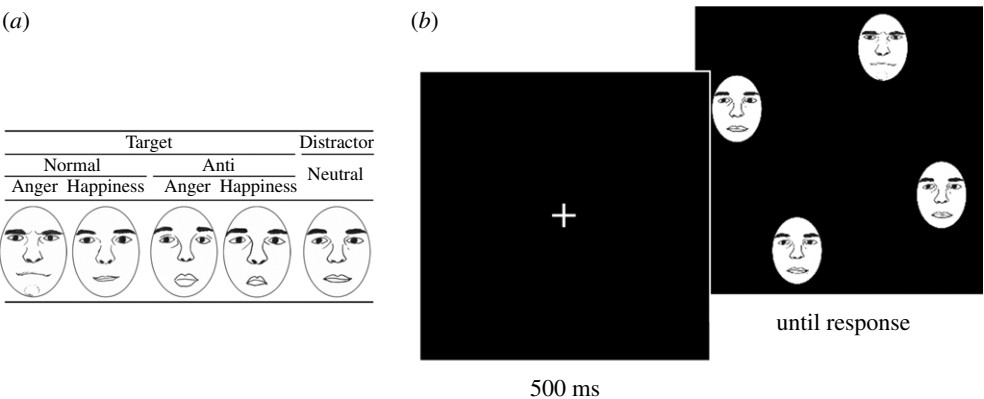

(a)

| | Target | | | | Distractor |
|---|---|---|---|---|---|
| | Normal | | Anti | | Neutral |
| | Anger | Happiness | Anger | Happiness | |

(b)

500 ms

until response

**Figure 1.** (*a*) Illustration of normal and anti-expression stimuli. (*b*) A trial sequence of the emotion–detection task. Published under the terms of the CC-BY (https://doi.org/10.1038/s41598-021-99945-y).

## 2.3. Procedure

### 2.3.1. Emotion–detection task

The emotion–detection task was the same as that used in a previous study [33,35]. First, a white cross was presented for 500 ms and then a stimulus array consisting of four faces was presented until the participant responded (figure 1). Participants were asked to answer whether all faces were the same or if one face was morphologically different as quickly and accurately as possible by pushing buttons using their left or right index finger. Normal emotional faces (angry or happy) obtained from two Caucasian models or their anti-expression faces were presented as a target in an array including three neutral faces as the distracters.

Anti-expressions were created using computer-morphing software (FUTON System, ATR) on a Linux computer. The coordinates of 79 facial feature points were identified manually and realigned based on the coordinates of the bilateral irises. Next, the differences between the feature points of the emotional (angry and happy) and neutral facial expressions were calculated. The positions of the feature points for the anti-expressions were determined by moving each point the same distance in the direction opposite to that in the normal emotional expressions. Minor colour adjustments were performed using Photoshop 5.0 (Adobe). Two types of additional adjustments were made to the stimuli. First, the photographs were cropped into a circle to eliminate contours and hair. Second, significant differences in contrast were eliminated. As a result, anti-expressions were morphologically reversed facial expressions of normal expression. Anti-expressions contained equivalent visual changes in facial features (e.g. eyebrows and lips) as normal emotional expressions of anger and happiness compared with neutral faces; however, the anti-expressions were recognized more frequently as neutral expressions than other emotions [31]. Representative images of anti-expressions are shown in figure 1 (also see [35]).

Each face image (2.5° wide × 3.5° high) occupied one of eight possible positions. The possible positions were equidistant on a circle with a radius of 14.0°; they were located at two coordinates from the centre ($x = 5.4°$, $y = 12.9°$ and $x = 12.9°$, $y = 5.4°$) and at their symmetrical positions with respect to the horizontal and vertical line. The eccentricity was within a range in which people can detect facial emotion (cf. [40]). The assignment of the response buttons was counterbalanced across participants, and the participants were asked to fixate on the cross during each trial. The experiment consisted of 432 trials presented in six blocks of 72 trials with an equal number of target-present and target-absent trials. The trial order was pseudo-randomized and each participant conducted 20 practice trials.

### 2.3.2. Rating task

Participants evaluated each stimulus used in the emotion–detection task in terms of valence and arousal (i.e. the nature and intensity of emotion, respectively, that the participant felt when perceiving the stimulus) using a nine-point scale (1 [negative and low arousal, respectively] to 9 [positive and high arousal, respectively]). The participants also rated the familiarity and naturalness of each stimulus. To illustrate that the anti-expressions were rated as less emotional than the normal expressions, only the results of the valence and arousal ratings are reported in the main text (see electronic supplementary

**Table 1.** Results of the emotion–detection task. $n = 64$, RT: reaction time, s.d.: standard deviation.

| | | RT | | accuracy | |
|---|---|---|---|---|---|
| | | mean | s.d. | mean | s.d. |
| normal | anger | 1287.9 | 353.7 | 91.0 | 6.9 |
| | happiness | 1422.6 | 374.3 | 87.6 | 11.1 |
| anti | anger | 1560.2 | 445.0 | 71.8 | 17.2 |
| | happiness | 1558.0 | 489.2 | 71.8 | 17.4 |

material file for other ratings). The order of stimuli and rating contents were randomized and balanced across participants.

### 2.3.3. Emotion–recognition task

The emotion–recognition task used in the present study was similar to that used in previous studies [41–43]. A total of 48 photographs of faces expressing six basic emotions (anger, disgust, fear, happiness, sadness or surprise) from four Caucasian and four Asian models were selected from photograph sets [44,45]. In each trial, a face photograph was presented on the centre of the display and written labels of the six basic emotions were presented around the photograph. To avoid any confusion regarding the buttons assigned to the six emotion labels, participants were required to answer which of the labels of the emotions best indicated the expressed emotion. Participants were instructed to carefully consider all labels for each face photograph. To investigate the deliberation process used to identify other people's emotions, no time limits were set in the emotion–recognition task. Thus, the labels remained on the screen until a verbal response was obtained. An experimenter recorded the verbal response using six buttons on a keyboard assigned to each emotion label. Feedback was not provided. The task consisted of Caucasian and Asian face blocks of 24 trials. The participants completed a total of 48 trials. The order of the blocks was counterbalanced across participants. The trial order was randomized within each block for each participant. Two practice trials were performed prior to the experiment.

## 2.4. Data analysis

SPSS 16.0 J software (SPSS Japan, Tokyo, Japan) was used to perform statistical analyses.

### 2.4.1. Emotion–detection task

The mean RTs in correct trials and accuracy under the normal expression and anti-expression conditions were calculated for each participant. Five participants with low correct response rates (less than 50%) and one participant with a difficulty of discriminating anti-expressions and neutral expressions were excluded from all further analyses of the emotion–detection task and rating task, and trials with RTs ± 3 s.d. from the mean were excluded from the RT analyses. The detection accuracy in the remaining participants was sufficiently high ($M = 80.5\%$, s.d. $= 11.0$) and there was no evidence of a speed–accuracy trade-off phenomenon (table 1). Thus, only the results of the RT were analysed as in previous studies [32–35], using a repeated-measures analysis of variance (ANOVA) with emotion (anger and happiness) and expression type (normal expression and anti-expression) as within-participant factors. When a significant interaction was found, the simple effects of expression type and emotion were analysed.

### 2.4.2. Rating task

Each rating was analysed using the same model as that used to analyse the mean RT; one additional participant was excluded from the analyses due to a technical problem with data acquisition.

### 2.4.3. Emotion–recognition task

Total accuracy (%) across the six basic emotions was calculated for each participant.

**Table 2.** Mean and s.d. for each condition in the rating tasks. $n = 63$, s.d.: standard deviation.

| | | valence | | arousal | |
|---|---|---|---|---|---|
| | | mean | s.d. | mean | s.d. |
| normal | anger | 2.4 | 0.9 | 7.0 | 1.3 |
| | happiness | 7.2 | 0.8 | 6.1 | 1.0 |
| anti | anger | 4.7 | 0.8 | 4.5 | 1.3 |
| | happiness | 3.8 | 0.8 | 4.2 | 1.2 |

### 2.4.4. Relationships between task performances and schizotypy

Two-step hierarchical linear regression analyses with each task performance as a dependent variable were conducted. The first step included age, estimated IQ and sex as independent variables. The second step included the SPQ score (total or each subscale score) to confirm whether the degree of schizotypy would have a significant association with task performance even when the effects of IQ, sex and age were controlled for. For the emotion–detection task, we used the ratios of the RT under anti-expressions to normal expressions condition as an index of effective detection for normal versus anti-expressions. The index could eliminate the influence of individual differences in response speed. A higher score was indicative of a more effective detection of normal expressions versus anti-expressions. For the emotion–recognition task, total accuracy was used as an index of task performance. $p$-values $< 0.05$ were considered to indicate statistical significance.

# 3. Results

## 3.1. Emotion–detection task

The results of the mean RT and the per cent accuracy are shown in table 1. The ANOVA of the mean RT revealed significant main effects of type, $F_{1,63} = 56.559$, $p < 0.001$, $\eta_p^2 = 0.473$, and emotion, $F_{1,63} = 16.633$, $p < 0.001$, $\eta_p^2 = 0.209$. These main effects were qualified by a significant interaction between expression type and emotion, $F_{1,63} = 9.239$, $p = 0.003$, $\eta_p^2 = 0.128$. *Post hoc* analyses of the interaction revealed significant simple effects of expression type for anger, $F_{1,63} = 144.834$, $p < 0.001$, $\eta_p^2 = 0.697$, and happiness, $F_{1,63} = 9.292$, $p = 0.003$, $\eta_p^2 = 0.129$. There were shorter RTs for normal expressions than anti-expressions for anger and happiness conditions (table 1). A significant simple effect of emotion was also found for normal expressions, $F_{1,63} = 18.687$, $p < 0.001$, $\eta_p^2 = 0.229$, but not anti-expressions, $F_{1,63} = 0.009$, $p = 0.926$, $\eta_p^2 < 0.001$. There were shorter RTs for anger than for happiness under the normal expression condition.

## 3.2. Rating task

The results for valence and arousal rating are shown in table 2.

### 3.2.1. Valence

The ANOVA revealed significant main effects of expression type, $F_{1,62} = 25.692$, $p < 0.001$, $\eta_p^2 = 0.293$, and emotion, $F_{1,62} = 415.773$, $p < 0.001$, $\eta_p^2 = 0.870$. These main effects were qualified by a significant interaction between expression type and emotion, $F_{1,62} = 614.625$, $p < 0.001$, $\eta_p^2 = 0.908$. *Post hoc* analyses found significant simple main effects of expression type for anger, $F_{1,62} = 222.212$, $p < 0.001$, $\eta_p^2 = 0.782$, and happiness, $F_{1,62} = 501.663$, $p < 0.001$, $\eta_p^2 = 0.890$. The participants evaluated the anti-anger expression as less negative than the normal-anger expression, as well as rated the anti-happiness expression as less positive than the normal-happiness expression. The results suggest that participants experienced anti-expressions as more emotionally neutral than normal expressions. There were also simple main effects of emotion for normal expressions, $F_{1,62} = 677.470$, $p < 0.001$, $\eta_p^2 = 0.916$ and anti-expressions, $F_{1,62} = 75.197$, $p < 0.001$, $\eta_p^2 = 0.548$. The participants evaluated normal-anger

**Table 3.** Two-step hierarchical regression analysis of the emotion–detection task ($n = 64$). IQ: intelligence quotient, SPQ: Schizotypal Personality Questionnaire, $\beta$: standardized beta, $R^2$: variance explained by the independent variable in the model.

| | step 1 | | | step 2 | | |
|---|---|---|---|---|---|---|
| | $\beta$ | $t$ | $p$-value | $\beta$ | $t$ | $p$-value |
| age | 0.020 | 0.164 | 0.870 | 0.021 | 0.177 | 0.860 |
| IQ | 0.266 | 1.951 | 0.056 | 0.247 | 1.858 | 0.068 |
| sex | −0.296 | −2.172 | 0.034 | −0.247 | −1.837 | 0.071 |
| total SPQ score | | | | −0.262 | −2.156 | 0.035 |
| | adjusted $R^2 = 0.046$ | | | adjusted $R^2 = 0.100$ | | |
| | $F_{3,63} = 2.003$, $p = 0.123$ | | | $\Delta F_{1,59} = 4.649$, $p = 0.035$ | | |

expressions as more negative than normal-happiness expressions and rated anti-anger expressions as more positive than anti-happiness expressions.

### 3.2.2. Arousal

We found significant main effects of expression type, $F_{1,62} = 231.583$, $p < 0.001$, $\eta_p^2 = 0.789$, and emotion, $F_{1,62} = 19.757$, $p < 0.001$, $\eta_p^2 = 0.242$. These main effects were qualified by a significant interaction between expression type and emotion, $F_{1,62} = 4.360$, $p = 0.041$, $\eta_p^2 = 0.066$. *Post hoc* analyses showed significant simple main effects of expression type for anger, $F_{1,62} = 130.773$, $p < 0.001$, $\eta_p^2 = 0.678$, and happiness, $F_{1,62} = 89.450$, $p < 0.001$, $\eta_p^2 = 0.591$. The participants gave a lower arousal rating for anti-expressions than for normal expressions, irrespective of the emotion category. There was also a simple main effect of emotion for normal expressions, $F_{1,62} = 16.104$, $p < 0.001$, $\eta_p^2 = 0.206$, but not for anti-expressions, $F_{1,62} = 2.083$, $p = 0.154$, $\eta_p^2 = 0.033$. The participants gave higher arousal rating for the normal-anger expression than for the normal-happiness expression.

### 3.3. Emotion–recognition task

The per cent accuracy across six basic emotions was $M = 71.0$, s.d. $= 9.2$.

### 3.4. Relationships between schizotypy and emotion detection

To examine whether schizotypy had an association with the performance for emotion detection after controlling for the effects of IQ, age and sex, two-step hierarchical linear regression analyses were conducted. There was a significant effect of sex ($\beta = -0.296$, $p = 0.034$) but not IQ and age in the first step (table 3). Female participants more effectively detected normal versus anti-expressions than did male participants. When total SPQ score was added to the regression model in the second step, a significant additional effect was observed, $F$-change$_{1,59} = 4.649$, $p = 0.035$. Overall schizotypy showed a significant negative correlation with the effectiveness of detecting normal versus anti-expressions ($\beta = -0.262$, $p = 0.035$; figure 2). When the score of disorganized or positive schizotypy was added to the model on behalf of the total SPQ score, a significant effect was also observed (disorganized: $F$-change[1, 59] = 11.676, $p = 0.001$; positive: $F$-change[1, 59] = 4.439, $p = 0.039$). The degree of disorganized ($\beta = -0.392$, $p = 0.001$) and positive schizotypy ($\beta = -0.253$, $p = 0.039$) showed a significant negative correlation with the effectiveness of detecting normal versus anti-expressions. When the score of negative schizotypy was added to the model on behalf of the total score, we observed no significant additional effect ($F$-change[1, 59] = 0.555, $p = 0.459$).

### 3.5. Relationships between schizotypy and emotion recognition

Two-step hierarchical linear regression analyses showed no significant effects of IQ, age or sex (table 4) in the first step. However, a significant additional effect of the score of positive schizotypy was found, $F$-change$_{(1,58)} = 7.177$, $p = 0.009$. The degree of positive schizotypy had a significant negative correlation

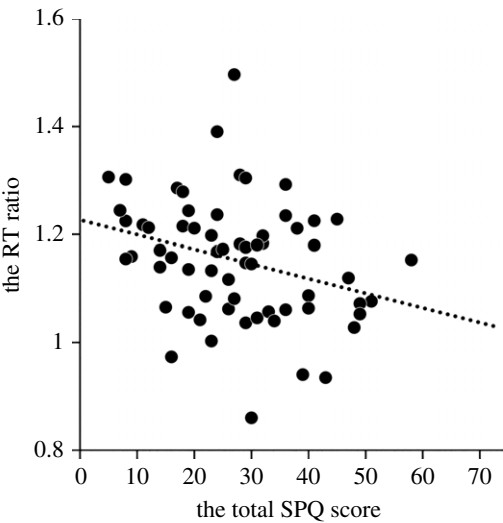

**Figure 2.** Scatterplots of the relationships between the reaction times (RTs) for the emotion–detection task and the total SPQ scores. The ratios of the RT under anti-expressions to normal expressions condition were negatively correlated with the scores.

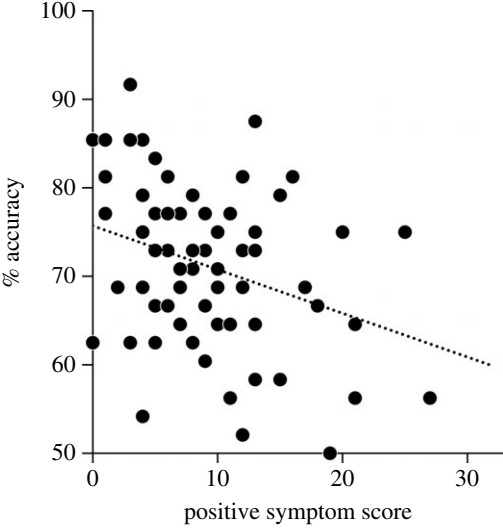

**Figure 3.** Scatterplot of the relationship between accuracy under the emotion–recognition task and the score of positive schizotypy.

**Table 4.** Two-step hierarchical regression analysis of the emotion–recognition task ($n = 70$). IQ: intelligence quotient, $\beta$: standardized beta, $R^2$: variance explained by the independent variable in the model.

|  | step 1 | | | step 2 | | |
|---|---|---|---|---|---|---|
|  | $\beta$ | $t$ | $p$-value | $\beta$ | $t$ | $p$-value |
| age | −0.065 | −0.539 | 0.592 | −0.073 | −0.638 | 0.526 |
| IQ | 0.157 | 1.174 | 0.245 | 0.151 | 1.181 | 0.242 |
| sex | −0.209 | −1.561 | 0.123 | −0.188 | −1.467 | 0.147 |
| positive symptoms score |  |  |  | −0.309 | −2.679 | 0.009 |
|  | adjusted $R^2 = 0.000$ | | | adjusted $R^2 = 0.086$ | | |
|  | $F_{3,66} = 1.007, p = 0.395$ | | | $\Delta F_{1,65} = 7.177, p = 0.009$ | | |

with the performance under the emotion–recognition task ($\beta = -0.309$, $p = 0.009$; figure 3). When the score of total, negative or disorganized schizotypy was added to the regression model on behalf of the score of positive schizotypy, we observed no significant additional effects of these scores (total: $F$-$change$[1, 59] = 1.141, $p = 0.289$; negative: $F$-$change$[1, 59] = 0.077, $p = 0.783$; disorganized: $F$-$change$[1, 59] = 0.600, $p = 0.441$).

# 4. Discussion

The results of the emotion–detection task replicated those of previous studies [32–35]. Normal expressions of anger and happiness were detected more rapidly and accurately than their anti-expressions, and anger was more rapidly detected than happiness in normal expressions. The rating tasks revealed that the anti-expressions of anger and happiness were rated as less negative and positive, respectively, than their normal expressions and that the anti-expressions were rated as less arousing than normal expressions. These results suggest that the rapid detection of normal expressions compared with anti-expressions generally depends on the high emotionality triggered by normal expressions. Performances on the emotion–recognition task were within range of those observed in Japanese adults [41–43]. These results indicate that the performances on both tasks were suitable for evaluating their relationships with the degree of schizotypy.

The degree of overall schizotypy was negatively correlated with effectiveness for detecting normal versus anti-expressions, even when controlling for IQ, age and sex. The results suggest that people with high levels of schizotypy have difficulties in detecting emotional facial expressions more rapidly than anti-expressions. A recent study also showed a lower accuracy for detecting emotional faces in patients with schizophrenia than controls [30]. However, a previous study reported the rapid detection of angry versus happy faces in both patients and controls [29]. These studies did not investigate detection performances using emotionally neutral faces presented randomly in a crowd of neutral faces, and the investigation of clinical populations might be affected by the differences in various clinical conditions. We investigated the relationship between facial expression processing and schizotypy in the general population and suggest that the detection of emotional facial expressions is atypical in people with high levels of schizotypy. Given that previous studies have demonstrated that emotional facial expressions were rapidly detected because of their emotional significance [32,35], the present findings also suggest that individual differences in schizotypy are associated with those of the perceptual as well as emotional processing of facial expressions.

Interestingly, although anti-expressions were generally rated as less arousal and less valence than normal emotional expressions, our findings suggest that people with high levels of schizotypy were less likely to have a priority to process normal emotional facial expressions compared with their anti-expressions. A meta-analysis demonstrated that people with schizophrenia tend to be more aroused by neutral stimuli [46] and show hyperactivations in response to neutral faces in brain regions associated with emotion processing (e.g. the amygdala) [47]. An electrophysiological study found that patients exhibit atypically heightened event-related potentials in response to neutral faces but not to emotional faces [48]. Hence, patients with schizophrenia may be sensitive to faces carrying ambiguous emotional meaning. These findings suggest that people with high levels of schizotypy also process not only normal but also anti-expressions differently compared with people with low schizotypy, and the small emotional differences between anti- and normal expressions may hinder the preferential detection of facial expressions carrying significant emotional meanings (e.g. normal anger and happiness).

Similar to the total SPQ score, the degrees of disorganized and positive schizotypy were negatively correlated with the effectiveness of detecting normal versus anti-expressions. These results might also support the view that people with high levels of schizotypy may not treat anti-expressions as relatively neutral stimuli. Individuals with disorganized and positive schizotypy exhibit high emotionality but less emotional clarity regarding their own emotions [49,50]. Previous studies have consistently demonstrated that emotional facial expressions are rapidly detected because of their emotional significance [32,35]. These findings suggest that people with high levels of disorganized or positive schizotypy are likely to be emotionally overwhelmed and might not be able to use their own emotional state as a cue to respond more rapidly to objects with biological significance, such as normal emotional expressions, compared with relatively neutral stimuli such as anti-expressions.

The degree of positive schizotypy was negatively correlated with the ability to recognize emotional facial expressions, consistent with previous findings showing that positive, as well as other, schizotypy are associated with impairments in emotion recognition in the general population [6]. Positive schizotypy might be associated with various domains of processing, because emotion recognition involves several processing stages [28]. Positive symptom domain consists of cognitive and perceptual alterations and paranoia [1,39]. The degree of perceptual aberration is negatively correlated with the ability to recognize emotion in people with schizophrenia, which suggests that atypical bodily experiences and an altered self–other boundary prevent the attribution of appropriate emotions to facial expressions

[51]. People with paranoid schizophrenia and those at high risk for psychosis have a tendency to recognize neutral faces as angry [52,53], which suggests that ambiguous facial expressions are likely to be mistaken for expressions that resemble emotional ones. Further studies investigating perceptual/emotional and attributional processing stages separately are needed to understand how positive schizotypy is associated with the ability of recognizing emotional facial expressions.

Although the present findings suggest the relationship between schizotypy and facial expression processing in the general population, they would provide an implication for our understanding of how facial-expression processing is impaired in people on the schizophrenia spectrum. Previous studies have consistently reported impairments in emotion recognition in patients with schizophrenia [6,9]. Likewise, a recent study observed impaired detection of emotional faces in patients with schizophrenia when they were explicitly referred to the emotional characteristics of a target (i.e. happy face [30]). The present study suggests the possibility that these findings are also found in people with schizophrenia under the emotion–detection task without explicit instructions regarding the emotional aspect of faces. A previous event-related potential study found that the mismatch-negativity response, which reflects the automatic processing of deviant emotional faces, is reduced in people with schizophrenia [54]. A meta-analysis study found that patients with schizophrenia show reduced activation in the amygdala in response to emotional facial expressions [27] involving the automatic processing of emotional significance [55]. Based on these findings, it is possible that not only attributional but also perceptual/emotion processing of facial expressions is atypical across people with high levels of schizotypy and patients with schizophrenia. However, in contrast with the present study, behavioural evidence suggests that the automatic perceptual/emotional processing of facial expressions are preserved in people with schizophrenia [19,20,22,23]. Further studies are needed to investigate whether people with schizophrenia have also difficulties detecting emotional faces under the emotion–detection task focusing on the automatic processing of emotional expressions. Additionally, given that the degree of positive and disorganized, but not negative schizotypy, was associated with the detection and/or recognition of facial emotions, each symptom domain might be differently associated with several aspects of facial expression processing. Further investigation of the relationship between each symptom domain and the processing of several types of facial expressions (e.g. emotional and non-emotional) would provide a clue to elucidate which aspects of facial expression processing are impaired and preserved in people with high levels of schizotypy and schizophrenia.

The present study has limitations that should be acknowledged. First, this study investigated the association of schizotypy with the processing of emotional facial expressions in the general population. This approach is advantageous because it can remove the influences of clinical conditions in patients and contribute to the understanding of emotion processing across clinical and general populations. However, further studies will be necessary to extend the current findings to the schizophrenia spectrum, because schizophrenia is a heterogeneous disorder and only a few studies have investigated emotion detection in clinical populations [29,30]. Second, other personality traits could potentially affect emotion detection, such as autistic traits [56] and neuroticism [34]. A recent study reported that individual differences in affect-related factors (anxiety, empathy and autistic traits) influence the performance in the emotion–recognition task [57]. To understand individual differences in cognitive function relevant to schizotypy, the effects of these traits and their interactions with schizotypy should be explored in future studies. Third, in contrast with previous studies [6,12–15], the results of the emotion–recognition task did not show a negative association with the degree of schizotypy other than positive schizotypy. To conclude that positive schizotypy specifically plays an important role in recognizing facial expressions, we need to conduct a further study with a larger sample size and measurements for assessing the above-mentioned affect-related factors.

## 5. Conclusion

The present study demonstrated that the degree of schizotypy was negatively correlated with the effective detection of normal versus anti-expressions and the ability to recognize facial expressions of six basic emotions. These findings suggest that people with high levels of schizotypy have difficulties in the processing of emotional facial expressions regardless of whether a demanded task requires perceptual/emotional or attributional stage of processing.

Consent to participate. All participants provided written informed consent prior to participation.
Ethics. The experiment was approved by the Kyoto University Graduate School and Faculty of Medicine, Kyoto University Hospital Ethics Committee (R0658).

Data accessibility. The data are provided in electronic supplementary material [58].

Authors' contributions. S.U., W.S., R.S., S.Y. and M.T. conceived and designed the experiments. S.U. and S.K. performed the experiments. S.U. and W.S. analysed the data and wrote the first draft of the manuscript, and all authors contributed to the writing of the manuscript.

Competing interests. The authors declare that they have no competing interests.

Funding. This work was supported by Grant-in-Aid for Young Scientists (B) and Grant-in-Aid for Scientific Research (C), Japan Society for the Promotion of Science (JP16K17360 and JP20K03478 to S.U.). The funding source had no involvement in study design; in the collection, analysis or interpretation of data; in the writing of the report; or in the decision to submit the article for publication.

Acknowledgement. We would like to thank Emi Yokoyama for her support in recruiting participants.

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
