## [Peer Review File · Royal Society Open Science]

Review History

RSOS-211145.R0 (Original submission)

Review form: Reviewer 1

Is the manuscript scientifically sound in its present form?

Yes

Are the interpretations and conclusions justified by the results?

Yes

Is the language acceptable?

Yes

Do you have any ethical concerns with this paper?

No

Have you any concerns about statistical analyses in this paper?

No

Recommendation?

Accept with minor revision (please list in comments)

Comments to the Author(s)

Many thanks for the opportunity to review this interesting study.

It is a well-conceived and well-conducted study that is also clearly communicated in the manuscript. Honestly, I think it is publishable in its current form, and I have seen many less coherent and exacting papers in the literature. Although the sample is somewhat limited in being restricted to young adults, it has clear strengths of terms of controlling for IQ and using anti-expressions.

I have only a few minor comments that might be worth considering.

The Introduction says "The study of schizotypy not only allows us to investigate the psychological mechanisms underlying schizophrenia pathology under cognitively demanding tasks, but also contributes to our understanding of general cognitive function, because schizotypal traits are also pronounced in the general population and are associated with difficulties related to specific aspects of human cognition"

This statement strikes me as a little bold given the evidence we have about the relationship between the two. I would suggest it is more accurate to say something along the lines that schizotypy allows us to investigate mechanisms associated with schizophrenia and related characteristics as they are distributed across the general population. I also wonder whether the authors might be a little more cautious in the Discussion about the potential relevance for understanding schizophrenia.

The paper reports a power analysis but it's not clear what sort of analysis this power analysis relates to. Given the extensive statistical analysis, I wonder whether this was a power analysis for a relatively simple effect, meaning it may have been under-powered for interactions. It would be useful therefore to clarify what analysis the power analysis was calculated for.

Typos and phrasing:

p14. "These results indicate that the performances on both tasks were suitable for evaluating their relationships with the degrees of schizotypy." Probably better as "the degree of schizotypy"

p14. "suggest that the detection of emotional facial expressions is atypical in people with the high schizotypy." probably better as "people with high levels of schizotypy"

p15. "The degree of overall schizotypy was negatively correlated with the effectiveness for detecting normal versus anti-expressions" probably better as "The degree of overall schizotypy was negatively correlated with effectiveness for detecting normal versus anti-expressions"

p15. "A recent study also showed the difficulties detecting emotional faces in patients for accuracy measure". Quite hard to make sense of this sentence so best rephrased I think.

p17. "We found that the degrees of positive and disorganised, but not negative schizotypy, are associate"

p18. "The present study has limitations and implications that should be acknowledged". Slightly awkwardly phrased. Perhaps better as ""The present study has limitations with subsequent implications that should be acknowledged"

In the service of transparency and collegiality, I prefer to sign my reviews.
Dr Vaughan Bell, UCL

Review form: Reviewer 2

Is the manuscript scientifically sound in its present form?

Yes

Are the interpretations and conclusions justified by the results?

Yes

Is the language acceptable?

Yes

Do you have any ethical concerns with this paper?

No

Have you any concerns about statistical analyses in this paper?

No

Recommendation?

Accept with minor revision (please list in comments)

Comments to the Author(s)

This is an interesting study contributing to the large literature on individual differences in processing / recognition of facial expressions of emotion.

I had only two suggestions.

First, it would be useful to discuss work on affective factors and emotion recognition, (e.g., <https://pubmed.ncbi.nlm.nih.gov/33047005/>), since these affective factors might covary with schizotypy.

Second, a little more justification for the specific tasks used here would be helpful. Specifically, what are the advantages of including anti-faces? I can see the advantage of using these in studies of, e.g., visual adaptation and aftereffects, but it would be helpful for the authors to elaborate on their suitability when studying emotion recognition etc.

Decision letter (RSOS-211322.R0)

Dear Dr Uono

On behalf of the Editors, we are pleased to inform you that your Manuscript RSOS-211322 "Schizotypy is associated with difficulties detecting emotional facial expressions" has been accepted for publication in Royal Society Open Science subject to minor revision in accordance with the referees' reports. Please find the referees' comments along with any feedback from the Editors below my signature.

Please submit your revised manuscript and required files (see below) no later than 7 days from today's (ie 18-Oct-2021) date. Note: the ScholarOne system will 'lock' if submission of the revision is attempted 7 or more days after the deadline. If you do not think you will be able to meet this deadline please contact the editorial office immediately.

on behalf of Dr Oliver Robinson (Associate Editor) and Essi Viding (Subject Editor)
openscience@royalsociety.org

Associate Editor Comments to Author (Dr Oliver Robinson):

Associate Editor: 1

Comments to the Author:

Thank you for submitting your paper to RSOS. As you can see, the reviewers were generally very positive about the paper and we would be happy to accept the paper after you complete the very minor revisions suggested. Please respond to all the comments, but please pay specific attention to:

- toning down language which suggests that the study has broader relevance beyond the links between this specific task and schizotypy
- clarifying what specific test the study was powered for (and highlighting in the limitations any underpowered analyses)
- providing a little more detail about the study design decisions

Reviewer comments to Author:

Reviewer: 1

Comments to the Author(s)

Many thanks for the opportunity to review this interesting study.

It is a well-conceived and well-conducted study that is also clearly communicated in the manuscript. Honestly, I think it is publishable in its current form, and I have seen many less coherent and exacting papers in the literature. Although the sample is somewhat limited in being restricted to young adults, it has clear strengths of terms of controlling for IQ and using anti-expressions.

I have only a few minor comments that might be worth considering.

The Introduction says "The study of schizotypy not only allows us to investigate the psychological mechanisms underlying schizophrenia pathology under cognitively demanding tasks, but also contributes to our understanding of general cognitive function, because schizotypal traits are also pronounced in the general population and are associated with difficulties related to specific aspects of human cognition"

This statement strikes me as a little bold given the evidence we have about the relationship between the two. I would suggest it is more accurate to say something along the lines that schizotypy allows us to investigate mechanisms associated with schizophrenia and related characteristics as they are distributed across the general population. I also wonder whether the authors might be a little more cautious in the Discussion about the potential relevance for understanding schizophrenia.

The paper reports a power analysis but it's not clear what sort of analysis this power analysis relates to. Given the extensive statistical analysis, I wonder whether this was a power analysis for a relatively simple effect, meaning it may have been under-powered for interactions. It would be useful therefore to clarify what analysis the power analysis was calculated for.

Typos and phrasing:

p14. "These results indicate that the performances on both tasks were suitable for evaluating their relationships with the degrees of schizotypy." Probably better as "the degree of schizotypy"

p14. "suggest that the detection of emotional facial expressions is atypical in people with the high schizotypy." probably better as "people with high levels of schizotypy"

p15. "The degree of overall schizotypy was negatively correlated with the effectiveness for detecting normal versus anti-expressions" probably better as "The degree of overall schizotypy was negatively correlated with effectiveness for detecting normal versus anti-expressions"

p15. "A recent study also showed the difficulties detecting emotional faces in patients for accuracy measure". Quite hard to make sense of this sentence so best rephrased I think.

p17. "We found that the degrees of positive and disorganised, but not negative schizotypy, are associate"

p18. "The present study has limitations and implications that should be acknowledged". Slightly awkwardly phrased. Perhaps better as ""The present study has limitations with subsequent implications that should be acknowledged"

In the service of transparency and collegiality, I prefer to sign my reviews.
Dr Vaughan Bell, UCL

Reviewer: 2

Comments to the Author(s)

This is an interesting study contributing to the large literature on individual differences in processing / recognition of facial expressions of emotion.

I had only two suggestions.

First, it would be useful to discuss work on affective factors and emotion recognition, (e.g., <https://pubmed.ncbi.nlm.nih.gov/33047005/>), since these affective factors might covary with schizotypy.

Second, a little more justification for the specific tasks used here would be helpful. Specifically, what are the advantages of including anti-faces? I can see the advantage of using these in studies of, e.g., visual adaptation and aftereffects, but it would be helpful for the authors to elaborate on their suitability when studying emotion recognition etc.

===PREPARING YOUR MANUSCRIPT===

one version should clearly identify all the changes that have been made (for instance, in coloured highlight, in bold text, or tracked changes);

===PREPARING YOUR REVISION IN SCHOLARONE===

To revise your manuscript, log into <https://mc.manuscriptcentral.com/rsos> and enter your Author Centre - this may be accessed by clicking on "Author" in the dark toolbar at the top of the

page (just below the journal name). You will find your manuscript listed under "Manuscripts with Decisions". Under "Actions", click on "Create a Revision".

-- If you are requesting an article processing charge waiver, you must select the relevant waiver option (if requesting a discretionary waiver, the form should have been uploaded, see 'File upload' above).

-- If you have uploaded any electronic supplementary (ESM) files, please ensure you follow the guidance at <https://royalsociety.org/journals/authors/author-guidelines/#supplementary-material> to include a suitable title and informative caption. An example of appropriate titling and captioning may be found at https://figshare.com/articles/Table_S2_from_Is_there_a_trade-off_between_peak_performance_and_performance_breadth_across_temperatures_for_aerobic_scope_in_teleost_fishes_/3843624.

At the 'Review & submit' step, you must view the PDF proof of the manuscript before you will be able to submit the revision. Note: if any parts of the electronic submission form have not been

completed, these will be noted by red message boxes - you will need to resolve these errors before you can submit the revision.

Author's Response to Decision Letter for (RSOS-211322.R0)

See Appendix A.

Decision letter (RSOS-211322.R1)

Dear Dr Uono,

I am pleased to inform you that your manuscript entitled "Schizotypy is associated with difficulties detecting emotional facial expressions" is now accepted for publication in Royal Society Open Science.

Kind regards,
Royal Society Open Science Editorial Office
Royal Society Open Science

on behalf of Dr Oliver Robinson (Associate Editor) and Essi Viding (Subject Editor)
openscience@royalsociety.org

Appendix A

Dear Dr. Robinson,

Thank you for your email in which you advised us to revise our manuscript. We have revised the manuscript according to your comments. All comments and suggestions were very helpful to the revision of our manuscript. I appreciate the effort and time devoted for reviewing our manuscript.

Yours sincerely,
Shota Uono

Associate Editor 1

Thank you for submitting your paper to RSOS. As you can see, the reviewers were generally very positive about the paper and we would be happy to accept the paper after you complete the very minor revisions suggested. Please respond to all the comments, but please pay specific attention to:

- toning down language which suggests that the study has broader relevance beyond the links between this specific task and schizotypy
- clarifying what specific test the study was powered for (and highlighting in the limitations any underpowered analyses)
- providing a little more detail about the study design decisions

Response: Thank you for your helpful comments. We revised the manuscript with careful considerations according to your comments.

Reviewer 1

Point 1: The Introduction says "The study of schizotypy not only allows us to investigate the psychological mechanisms underlying schizophrenia pathology under cognitively demanding tasks, but also contributes to our understanding of general cognitive function, because schizotypal traits are also pronounced in the general population and are associated with difficulties related to specific aspects of human cognition"

This statement strikes me as a little bold given the evidence we have about the relationship between the two. I would suggest it is more accurate to say something along the lines that schizotypy allows us to investigate mechanisms associated with schizophrenia and related characteristics as they are distributed across the general population. I also wonder whether the authors might be a little more cautious in the Discussion about the potential relevance for understanding schizophrenia.

Response 1: Thanks for your helpful comments. According to the reviewer's comment, we modified the sentence in the introduction (Page 3) as follows. "The study of schizotypy allows us to investigate the psychological mechanisms associated with schizophrenia and related characteristics under cognitively demanding tasks, because schizotypal traits are distributed across the general population. Additionally, we modified the discussion about the implication of schizophrenia research in a careful manner (Page 17).

Point 2: The paper reports a power analysis but it's not clear what sort of analysis this power analysis relates to. Given the extensive statistical analysis, I wonder whether this was a power analysis for a relatively simple effect, meaning it may have been under-powered for interactions. It would be useful therefore to clarify what analysis the power analysis was calculated for.

Response 2: The sample size was determined through *a priori* power analysis for multiple regression analysis using G*power assuming the intermediate effect size ($r = 0.342$) obtained in the previous schizotypy study, α level of 0.05, and power of 0.8. The results showed that 62 participants would be needed. Considering the possibility that the RTs under the emotion-detection task cannot be calculated in some participants because of too low accuracy rate, a total of 70 Japanese adults were recruited. The resulting sample size was four times of the original emotion-detection study ($n = 17$) and was similar with the study ($n = 74$) investigating the relationship between personality trait and the emotion detection performance. Thus, we consider that the sample size is sufficient for detecting the effect of face type (normal and anti-expressions) under the emotion-detection task and its' relationship with the degree of schizotypy (Page 6). However, we also added this point as a limitation (Page 18), because contrast with previous schizotypy studies, our study did not find the association between the emotion-recognition ability and the degree of schizotypy other than positive schizotypy.

Point 3: Typos and phrasing

p14. "These results indicate that the performances on both tasks were suitable for evaluating their relationships with the degrees of schizotypy." Probably better as "the degree of schizotypy"

p14. "suggest that the detection of emotional facial expressions is atypical in people with the high schizotypy." probably better as "people with high levels of schizotypy"

p15. "The degree of overall schizotypy was negatively correlated with the effectiveness for detecting normal versus anti-expressions" probably better as "The degree of overall

schizotypy was negatively correlated with effectiveness for detecting normal versus anti-expressions"

p15. "A recent study also showed the difficulties detecting emotional faces in patients for accuracy measure". Quite hard to make sense of this sentence so best rephrased I think.

p17. "We found that the degrees of positive and disorganised, but not negative schizotypy, are associate"

p18. "The present study has limitations and implications that should be acknowledged". Slightly awkwardly phrased. Perhaps better as "The present study has limitations with subsequent implications that should be acknowledged"

Response 3: Thank you for your careful reading. I corrected the places that were pointed out.

Response to Reviewer 2

Point 1: First, it would be useful to discuss work on affective factors and emotion recognition, (e.g., <https://pubmed.ncbi.nlm.nih.gov/33047005/>), since these affective factors might covary with schizotypy.

Response 1: Thanks for your helpful suggestion. We cited the paper and addressed the importance of investigating affect-related factors in schizotypy study (Page 18).

Pont 2: Second, a little more justification for the specific tasks used here would be helpful. Specifically, what are the advantages of including anti-faces? I can see the advantage of using these in studies of, e.g., visual adaptation and aftereffects, but it would be helpful for the authors to elaborate on their suitability when studying emotion recognition etc.

Response 2: Anti-expressions were only included in the emotion-detection task. They contain a degree of visual change in facial features equivalent to those of normal emotional facial expressions relative to neutral facial expressions and are recognised as relatively neutral expressions compared to normal emotional expressions. Thus, the enhanced detection for normal versus anti-expressions can be attributable to the enhanced emotional processing rather than the visual difference between target faces. We added this sentence in the introduction (Page 6).